# Time Series Analysis of Cryptocurrency Prices Using Long Short-Term Memory

Jacques Phillipe Fleischer [1,*], Gregor von Laszewski [2,*], Carlos Theran [3] and Yohn Jairo Parra Bautista [3]

1 Kendall Campus, The Honors College at Miami Dade College, 11011 SW 104th St, Miami, FL 33176, USA
2 Biocomplexity Institute, University of Virginia, 994 Research Park Blvd, Charlottesville, VA 22911, USA
3 Computer & Information Systems Department, Florida A&M University, 1333 Wahnish Way 308 A Benjamin Banneker Technical Bldg, Tallahassee, FL 32307, USA; carlos.theran@famu.edu (C.T.); yohn.parrabautista@famu.edu (Y.J.P.B.)
* Correspondence: jacquespfleischer@gmail.com (J.P.F.); laszewski@gmail.com (G.v.L.)

**Abstract:** Digitization is changing our world, creating innovative finance channels and emerging technology such as cryptocurrencies, which are applications of blockchain technology. However, cryptocurrency price volatility is one of this technology's main trade-offs. In this paper, we explore a time series analysis using deep learning to study the volatility and to understand this behavior. We apply a long short-term memory model to learn the patterns within cryptocurrency close prices and to predict future prices. The proposed model learns from the close values. The performance of this model is evaluated using the root-mean-squared error and by comparing it to an ARIMA model.

**Keywords:** prediction; cryptocurrency; LSTM




## 1. Introduction

Cryptocurrencies, which reside on the blockchain, are a novel, new form of monetary value with their own ever-changing prices. Blockchain systems are decentralized, meaning cryptocurrency transactions are verified and available to all users for verification, transparency, and maximum accountability. This currency's technology is lucrative and becoming a sought-after investment as opposed to typical transactions through clearing houses and banks. Additionally, cryptocurrency is a revolutionary technology that could disrupt societal structures due to the anonymous nature of its transactions. Thus, due to its groundbreaking potential and the mania behind its investment potential, the currency's prices are extremely volatile. Bitcoin, in particular, has the highest price of all cryptocurrencies. As the coin's appraisal can change in thousands of dollars in a matter of a few days, the U.S. Securities and Exchange Commission advises that high risk accompanies these investments [1].

Artificial intelligence (AI) presents an effective way to predict future prices to counteract cryptocurrency coins' acute volatility, which can otherwise discourage venture capitalists from supporting a company that utilizes cryptocurrency [2]. Furthermore, the concept of pairing AI with economic trading creates an effective pairing that is an attractive business endeavor. For instance, a present-day, increasingly popular innovation is the automated trading of digital investment assets by AI, which is poised to become the norm of the future with very little human oversight [3]. The aforementioned automated trading of assets is not possible without the writing of a Python program that knows the best time to execute trades. Similarly, AI is applied in this experiment to predict the future price of cryptocurrencies on several different blockchains, including the Electro-Optical System and Ethereum.

Considering that the blockchain and cryptocurrency technologies are relatively novel, there is a need for research into analyzing cryptocurrency patterns and behavior. Previous research papers have explored currencies such as Bitcoin [4]. However, we seek to expand

on this research by analyzing other lesser-known coins such as the Electro-Optical System (EOS) token and Dogecoin. Researching these other technologies is vital as they serve different purposes; for example, the EOS and Ethereum coins are used to drive the operation of decentralized apps (DApps). If research explores the potential of this coin and finds it prospectively valuable, then it can drive the development and use of such apps for "gaming, finance, [or] social media" [5]. The creation of such apps can present a lucrative and convincing case for companies to consider. Thus, the goal of this paper is to create a model that can appraise the potential of a cryptocurrency through its historical close prices to find its economic potential. While the model is not meant to take the coin's purpose into consideration, it still has the potential to bring attention to a coin that has a high investment value. The research question is whether this can be attained through only analyzing the cryptocurrency's close price with a long short-term memory AI model.

Long short-term memory (LSTM) is a neural network, which is a form of AI, that ingests information and processes data using a gradient-based learning algorithm [6]. This algorithm's accuracy improves as it is provided more data. Thus, we propose that LSTMs are effective in predicting cryptocurrency price as the LSTM model can analyze pre-existing historical price data dating back many years. Additionally, the model can ideally output the predicted price in several timetables, such as a week, a month, or a year into the future. If successful, the predictions can assist in convincing venture capitalists to invest in cryptocurrency products with potentially high returns; the prediction can also entice investors into buying a coin that is poised to increase in price, which results in a price increase, which repeats. Thus, the main purpose of this research is to provide insight into any cryptocurrency coin and how its price will perform in the future. LSTM has been designed for forecasting and allows for many applications to time series analysis, such as COVID-19 prediction [7] or stock market prediction [8]. While both applications explore rapid changes and offer promising results, a question remains: this paper's research question asks: *Is LSTM an accurate model for predicting volatile cryptocurrency prices?*

## 2. Literature Review

One of the first studies to attempt to predict cryptocurrency using non-linear neural networks—conducted by Maiti, Vyklyuk, and Vukovic and published in 2020—found that the non-linear models were preferable to linear models as the former's error was lower than the latter, giving more accurate predictions. This study also found that some elements of the cryptocurrency's historical data were not helpful in prediction, such as the volume [9].

Another study released in 2022 by Critien, Gatt, and Ellul sought to expand the prediction beyond merely yielding a direction; instead of reporting whether the price would go up or down, the model would report "the magnitude of the price change" [10]. The study leveraged not only historical price data for Bitcoin, but also Twitter posts to glean public sentiment about the currency. An important distinction of this study is that it used a bidirectional LSTM, which is a model composed of two LSTMs: one for historical price data and one for Twitter posts. Its accuracy of 64.18% was achieved by using 450 days of historical data [10].

Furthermore, a 2022 study by Sarkodie, Ahmed, and Owusu used COVID-19 data, such as the number of cases and deaths, to determine whether cryptocurrencies such as Bitcoin would rise or fall in price. The study made use of the Romano–Wolf algorithm to test this correlation [11]. In particular, it deemed Bitcoin to be very highly correlated with deaths due to COVID-19, correspondingly fluctuating 90.50% of the time [11].

Lastly, a 2019 study by Azari utilized an autoregressive integrated moving average (ARIMA) approach to predict the price of Bitcoin [12]. In this study, the model's effectiveness was evaluated by examining the mean-squared error; the findings reported a very low level of prediction error [12].

Scientific papers such as these have proven that using artificial intelligence and neural networks yields accurate results in predicting cryptocurrency. However, as many parameters can be tweaked in this prediction goal (such as using volume data of the currencies,

using extraneous data such as COVID-19 statistics, or utilizing different forms of neural networks), we aim to determine whether it is possible to gain accurate predictions using only close price data and an LSTM. We also explore whether the LSTM is more accurate than the ARIMA model. Our research hypothesis is that the LSTM will provide a more accurate model, in that its predictions will produce a smaller root-mean-squared error (RMSE).

## 3. Datasets

This paper utilizes data obtained through the Python module *yfinance* [13], which downloads statistics regarding a stock ticker or cryptocurrency; the module's functions can be customized and executed with various parameters. We used the *period* parameter, set to max, to download historical prices of a cryptocurrency from the first day of its debut on the market to the present day when the program is executed (otherwise called the maximum period since it encapsulates the entire lifetime of the currency). Additionally, *yfinance* can provide different cryptocurrency data intervals, such as every minute, every five minutes, every hour, each day, and more. We also elected to use the one-day interval as we are predicting close prices: the cryptocurrency's price at the end of the day. The *yfinance* module uses data from Yahoo Finance [14], which is the source for Figure 1: a line graph of the EOS-USD cryptocurrency's close price. Figure 1 begins at 9 November 2017 because that is the day that EOS-USD debuted on the market.

While *yfinance* provides other ticker data such as *Open*, *High*, *Low*, and *Volume*, this paper does not discuss the use of such values in prediction—only the close values.

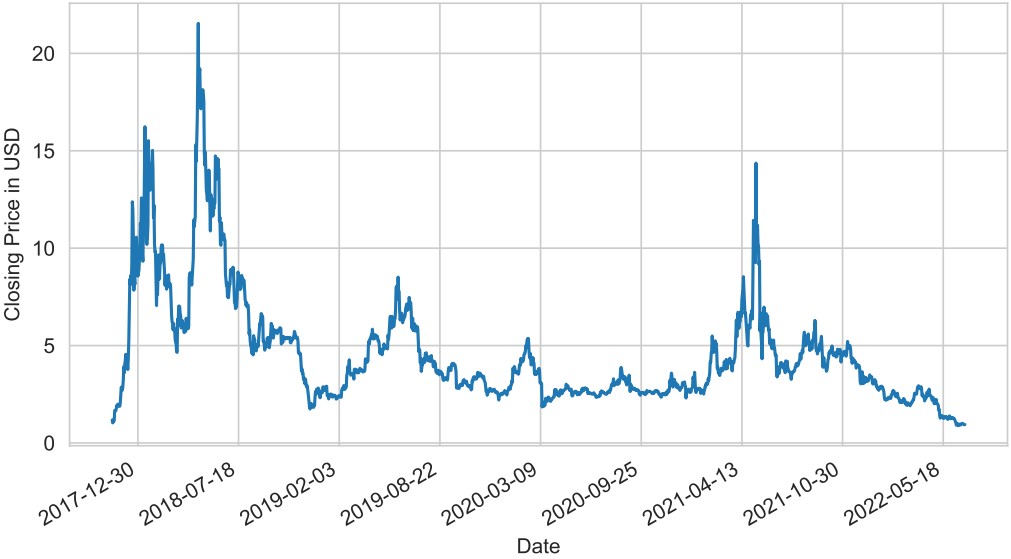

**Figure 1.** Line graph of EOS price from 9 November 2017 to 30 June 2022. Generated utilizing price data from the *yfinance* Python module [13], which scrapes from Yahoo Finance [14].

## 4. Architecture

Our architecture is based on the execution of four phases. The four phases are depicted in Figure 2. The phases are: (1) retrieving data from *yfinance*, (2) isolating the close prices from other data that are irrelevant to the prediction, (3) training the LSTM with historical close prices to predict future ones, and (4) plotting the prediction model, respectively.

Such an architecture, especially the incorporation of LSTM, is established and has been used numerous times in previous scientific literature [10,15–17]. We expand upon this preexisting work by using only the close prices; by conducting benchmarks in several computing environments; and by comparing the root-mean-squared error in correlation

with the number of epochs used during LSTM training. Lastly, we compare our LSTM model with an ARIMA model to gauge the accuracy and quality of the predictions.

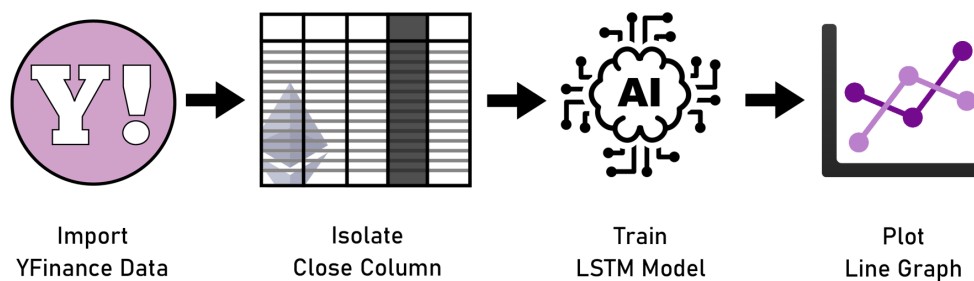

**Figure 2.** The process of producing a time series cryptocurrency prediction line graph using LSTM.

## 5. Implementation

A first approach for predicting cryptocurrency was to gain the close data by scraping financial websites. A Python module to initially achieve this was *BeautifulSoup*, but the module was not an ideal solution as the script had to be rewritten every time the financial websites' structure or layout changed. Additionally, to predict lesser-known cryptocurrencies such as EOS, we also explored Kaggle; however, the datasets provided there were not a viable option as they were deprecated and out of date. Thus, the most promising method we found was to download historical price data through the *yfinance* Python module, which returns the close values from the day of its first appearance on the market to the latest available price.

Our neural network framework of choice is the long short-term memory (LSTM) model because it has a memory capacity. As we are using a time series dataset, which means the data are historical and regularly occur over a period of time, the memory capacity is ideal. The model can remember historical patterns and use them to generate predictions [17].

The Jupyter notebook iterates through a Python function, which normalizes the data using min–max normalization; fits a long short-term memory model using Keras, a Python deep-learning API; undergoes model predictions; saves the figures of predictions; and outputs a log file with detailed benchmarks. The notebook can be altered to include other cryptocurrency tickers as long as they are available on Yahoo Finance. The notebook is also able to analyze traditional stock tickers.

The Python pseudocode for the Jupyter notebook program is shown in Figure 3.

The code contains multiple loops that iterate through the hyperparameters, including each cryptocurrency, each quantity of epochs, and the number of repeats so that the results can be presented in a statistically sound manner. The program also allows us to perform multiple runs and to select the best set of parameters for maximum accuracy. For more details about the program, the code on GitHub is available and open-source [18].

The notebook's first phase involves downloading the cryptocurrency's historical data using *yfinance* [13]. The data undergo normalization so that the model accuracy is easier to compare between different cryptocurrencies with widely different values. Next, the close data are isolated because this experiment only focuses on the close price. The close data were divided into a training set and test set, which were subsequently split into their own x and y sets for the purposes of the Keras LSTM model.

The learning model contains 250 units within its LSTM cell and is run through a dropout layer of 0.2 to prevent overfitting. Additionally, the model is run through a dense layer of 1 unit dimensionality to connect the neurons of the previous dropout layer. A diagram of the sequence of layers is shown in Figure 4. A visual description of the long short-term memory principle is showcased in [19].

LSTM Benchmark Program: *yfinance-lstm-all-figures.ipynb*

```
...
counter = 0
for each epoch in epochs:
  for each crypto in cryptos:
    get_crypto(crypto)
    for each i in range(repeat):
      lstm_crypto_benchmark(crypto,
                            id=i,
                            epoch=epoch)
      counter = counter + 1
    write_benchmark_to_log_file()
...
```

LSTM Benchmark Analysis Program: *yfinance-lstm-analysis-final.ipynb*

```
...
analyze_log_file()
...
```

**Figure 3.** Python pseudo-code to run the crypto benchmarks is split up into two parts: (1) to run the LSTM prediction and create the benchmark and (2) to run the analysis to create the figures.

Our architecture allows easy modification of the entire model with more layers, but also the adaptation of the hyperparameters.

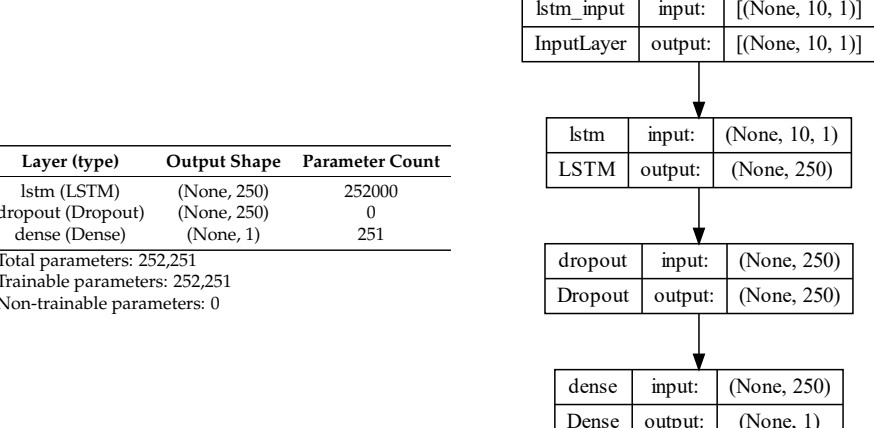

| Layer (type) | Output Shape | Parameter Count |
|---|---|---|
| lstm (LSTM) | (None, 250) | 252000 |
| dropout (Dropout) | (None, 250) | 0 |
| dense (Dense) | (None, 1) | 251 |

Total parameters: 252,251
Trainable parameters: 252,251
Non-trainable parameters: 0

**Figure 4.** Parameters of the LSTM model and the diagram of the sequential model, including the layers of LSTM.

Figures 5–7 use the EOS-USD dataset of close prices from 9 November 2017 to 30 June 2022. Within the trained model, only the 200 days at the end of the dataset are predicted so that the model learns from all of the prior days.

After the model was trained through 100 epochs, the Python notebook generated Figure 5, a line graph of the prediction model. Since it is hard to see the difference between the blue line (with the true close prices) and the red prediction line of the last 200 days, Figure 6 zooms into that point for human readability.

Figure 7 also shows the impact that epochs have on the accuracy of the prediction model. Figure 7 contains two graphed lines: a blue line representing the price of the EOS coin and a red line representing the model's prediction of the price. As the number of training epochs increases, the prediction becomes more and more accurate with respect to the actual price that the cryptocurrency was valued at on the market. In Figure 7, the green *history* line depicted in the legend is not shown because the graph is zoomed in to the later prediction phase, where the historical price data become the blue *true* line instead of green.

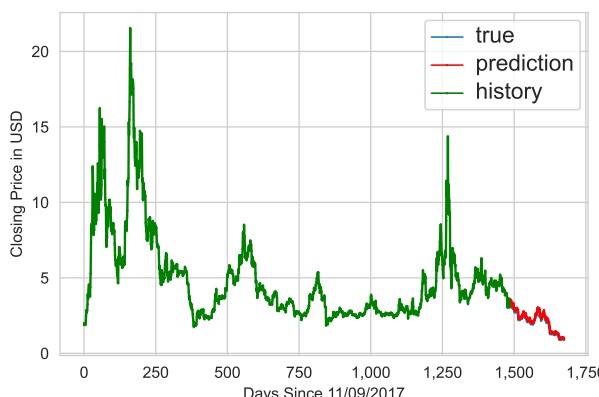

**Figure 5.** EOS-USD price overlayed with the latest 200 days predicted by LSTM ran through 100 epochs of training.

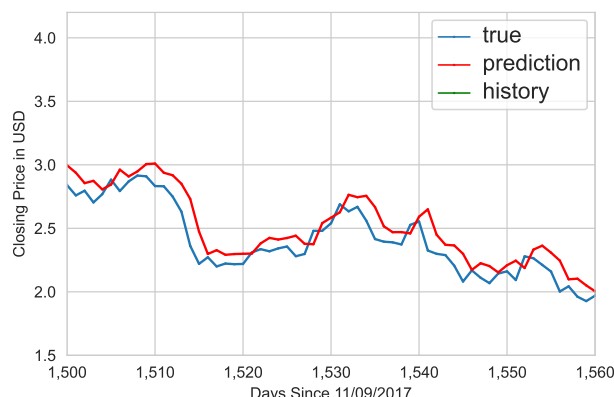

**Figure 6.** Zoomed-in graph for readability (same as Figure 5, but scaled x- and y-axis).

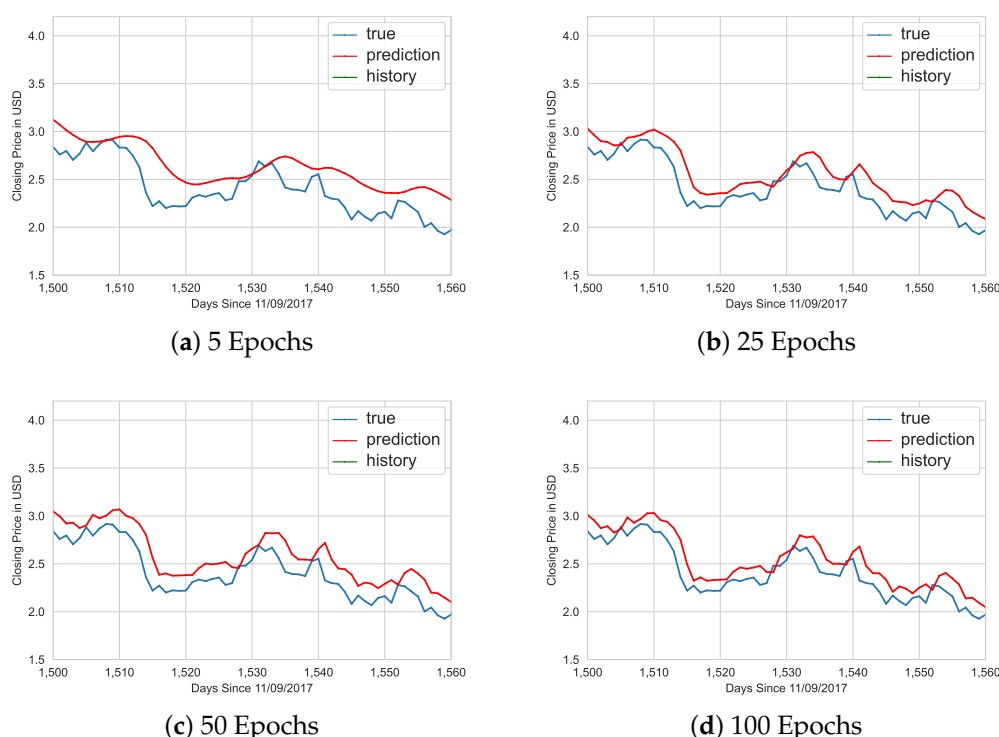

(**a**) 5 Epochs

(**b**) 25 Epochs

(**c**) 50 Epochs

(**d**) 100 Epochs

**Figure 7.** Effect of EOS-USD prediction model based on the number of epochs completed.

Lastly, cryptocurrencies other than EOS such as Bitcoin, Ethereum, and Dogecoin can be analyzed as well. Figure 8 demonstrates the prediction models generated for EOS and Bitcoin, whereas Figure 9 demonstrates the prediction models generated for Ethereum and Dogecoin.

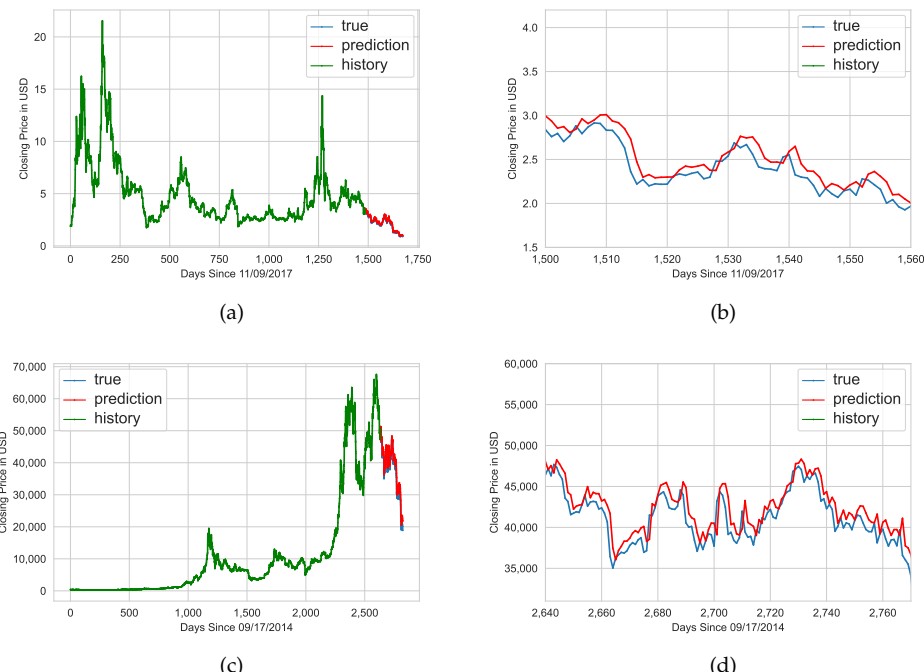

**Figure 8.** EOS and Bitcoin prediction models with 100 epochs of training. (**a**) EOS-USD close prices overlayed with prediction values. (**b**) Zoomed-in prediction values for EOS-USD. (**c**) BTC-USD close prices overlayed with prediction values. (**d**) Zoomed-in prediction values for BTC-USD.

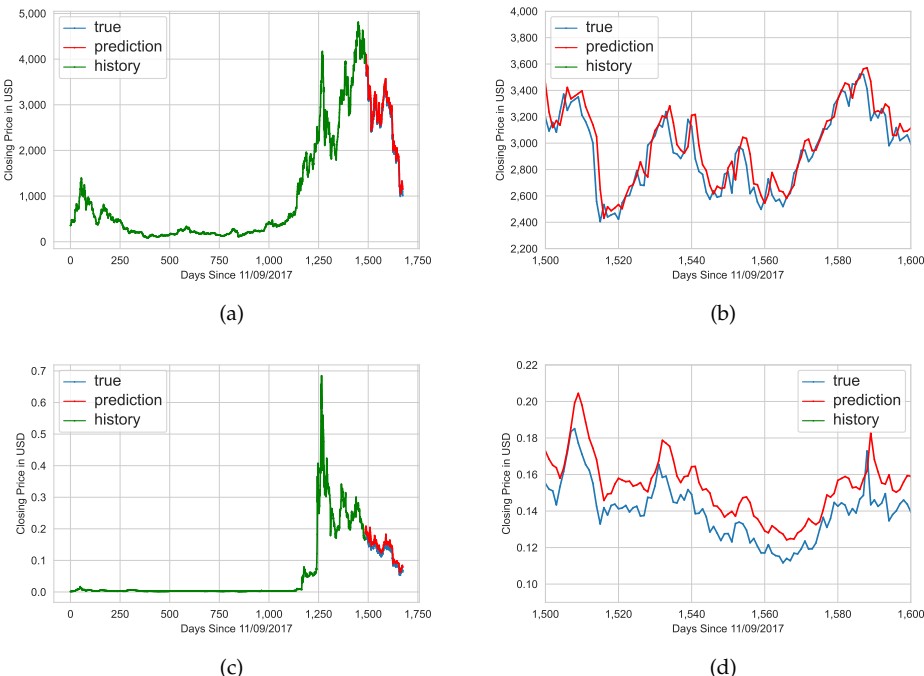

**Figure 9.** Ethereum and Dogecoin prediction models with 100 epochs of training. (**a**) ETH-USD close prices overlayed with prediction values. (**b**) Zoomed-in prediction values for ETH-USD. (**c**) DOGE-USD close prices overlayed with prediction values. (**d**) Zoomed-in prediction values for DOGE-USD.

The number of training epochs can affect the model's RMSE, which details how close the prediction line is to the actual close prices in United States Dollars (USD). As demonstrated in Figure 10, more epochs reduce the RMSE (but the change becomes negligible after around 100 epochs).

Figure 10 overlays each predicted cryptocurrency on a line graph after normalizing the RMSE values. The values are normalized since the cryptocurrencies have vastly different prices, but the normalization details the predictions' accuracies and evenly compares them after removing the artificial valuation.

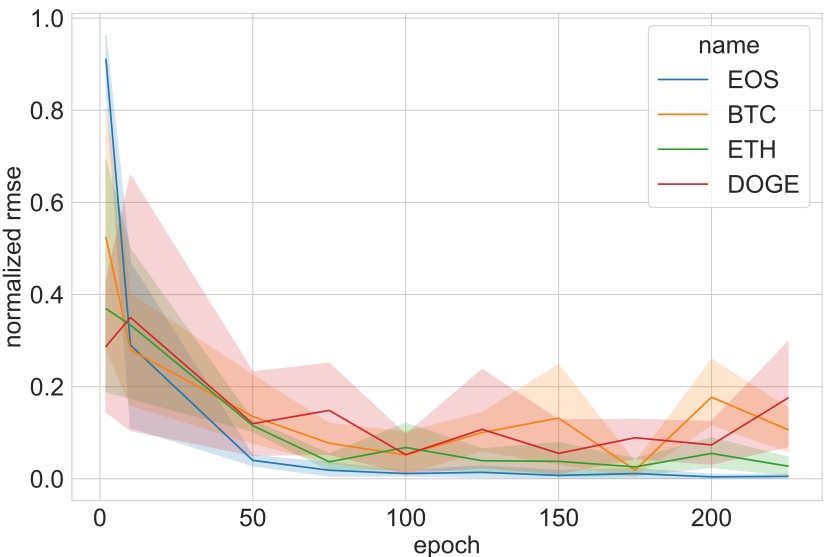

**Figure 10.** Normalized root-mean-squared error over number of training epochs.

## 6. Benchmark

The program was run on two computers (see Table 1). One computer was a 64-bit Windows 10 Home Edition (21H1) computer with an AMD Ryzen 5 3600 processor (3.6 GHz). Its memory was dual-channel 16 GB RAM clocked at 3200 MHz; its graphics card was a GTX 1660 Ventus XS OC. The other computer was a 64-bit Ubuntu 20.04.4 LTS with an AMD Ryzen 9 5950X processor (5.1 GHz), whose memory was 126 GB. The Ubuntu computer's graphics card was an RTX 3090. Table 1 lists the specifications of the Windows machine and the Ubuntu Linux machine, including the allocated computer memory during runtime and the Python version. The StopWatch module was used from the package *cloudmesh-common* [20] to print these specifications and to measure the training time.

**Table 1.** Ubuntu computer benchmark details regarding the specifications of the computer at the time of program execution.

| Attribute | Linux R9-5950X | Windows R5-3600 |
|---|---|---|
| CPU cores | 16 | 6 |
| CPU threads | 32 | 12 |
| CPU frequency | 5083.4 MHz | 3600.0 MHz |
| RAM available | 112.9 GiB | 2.3 GiB |
| RAM total | 125.7 GiB | 16.0 GiB |
| Python | 3.10.5 | 3.10.5 |
| Processor | AMD64 Ryzen 9 5950X | AMD64 Ryzen 5 3600 |
| OS | Ubuntu 20.04.1 | Windows 10.0.19043 |

Furthermore, Figures 11 and 12 plot the training time in seconds over number of epochs that the LSTM model was run through on the Windows and Ubuntu operating

systems, respectively. Similarly, Figures 13 and 14 plot the prediction time for Windows and Ubuntu, while Figures 15 and 16 plot the entire program runtime (training and prediction phases included) for Windows and Ubuntu. The runtime becomes much longer when training prediction models with many epochs; most of the time resides in the training portion, whereas the prediction portion only takes, at most, one second.

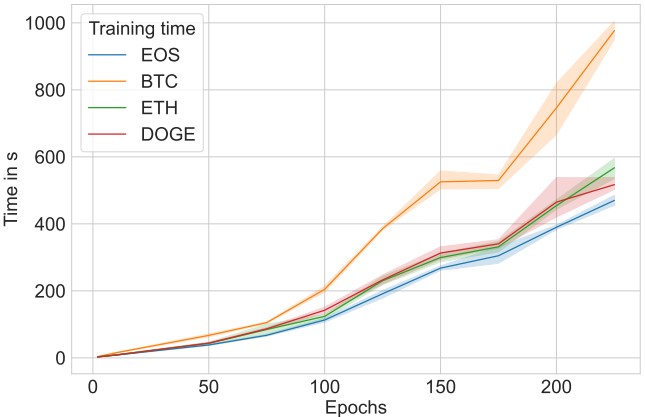

**Figure 11.** Windows training time over training epochs.

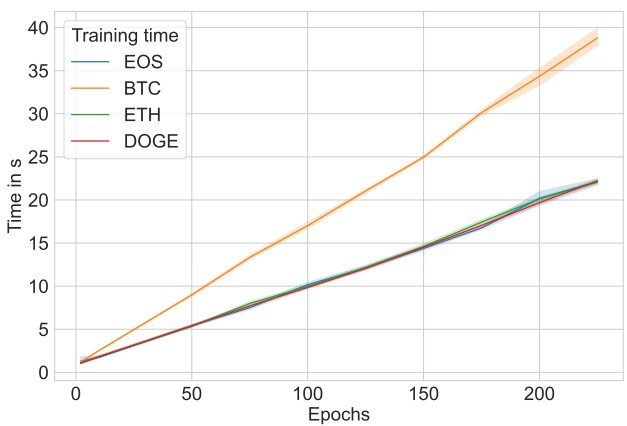

**Figure 12.** Ubuntu training time over training epochs.

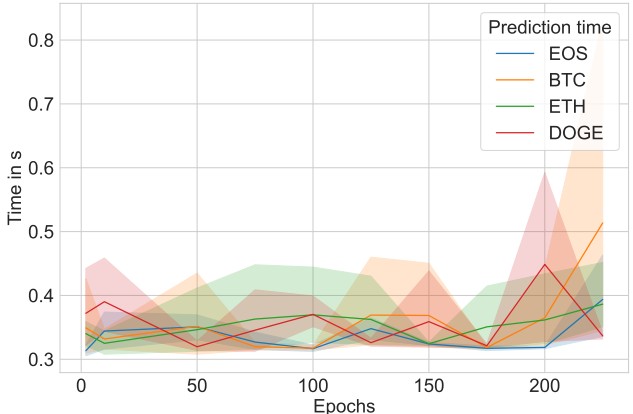

**Figure 13.** Windows prediction time over training epochs.

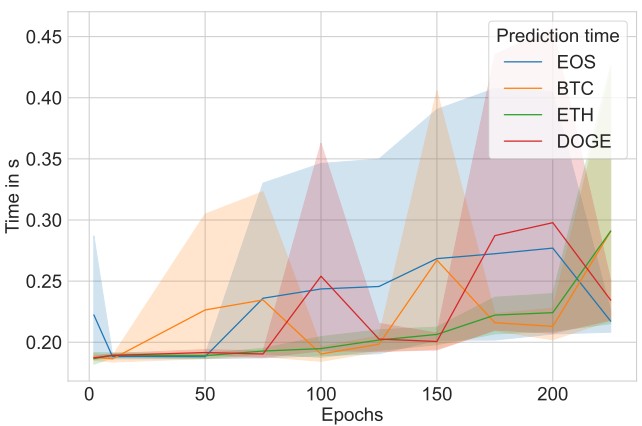

**Figure 14.** Ubuntu prediction time over training epochs.

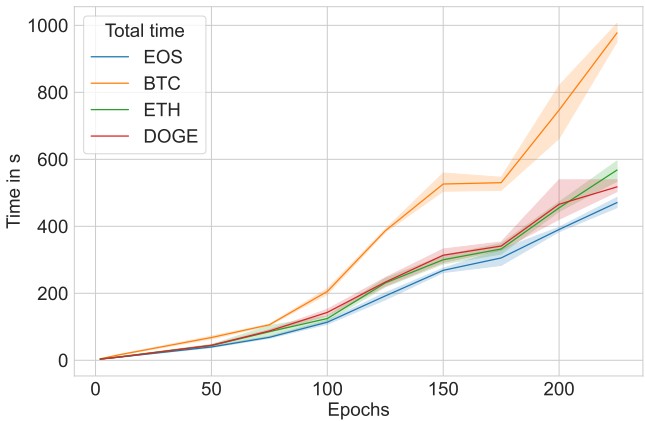

**Figure 15.** Windows total program runtime based on cryptocurrency and number of training epochs.

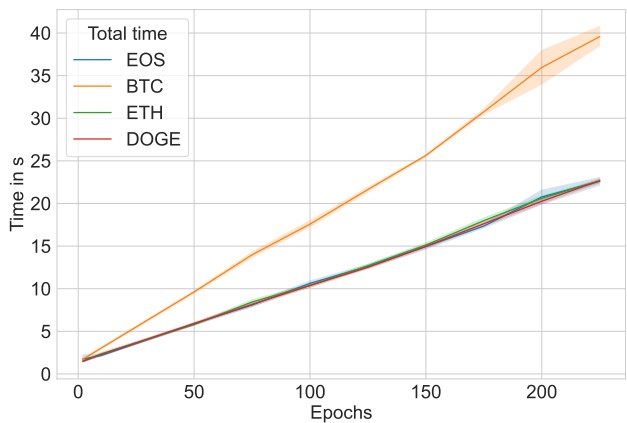

**Figure 16.** Ubuntu total program runtime based on cryptocurrency and number of training epochs.

## 7. Model Comparison

We compared the LSTM algorithm against an ARIMA model. For the ARIMA model, we used (p,d,q) = (5,1,0) for the autoregressive, differences, and moving average parameters, respectively, as they gave ideal results for the tested input data. For the LSTM models, we selected the model that resulted in the best fit. The comparison of the RMSE is shown in Table 2. We see that the best LSTM model we identified significantly outperformed the

ARIMA model. The difference is bigger for cryptocurrencies with overall smaller values as they have percentually larger fluctuations, which the LSTM algorithm seems to be able to handle better. Most importantly, the system is self-learning, and only the input data are used, so that there is no need to, for example, identify ARIMA parameters (p,d,q). This simplifies the model identification.

**Table 2.** RMSE model comparison.

| Cryptocurrency | LSTM RMSE | ARIMA RMSE | Improvement Using LSTM vs. ARIMA |
|---|---|---|---|
| EOS | 0.119 | 0.436 | +72.8% |
| BTC | 1334.755 | 1718.339 | +22.3% |
| ETH | 117.655 | 136.605 | +13.9% |
| DOGE | 0.007 | 0.025 | +72.0% |

In Table 3, we compare the average times of an individual optimization conducted either by LSTM or ARIMA on a particular dataset. As we can see, the ARIMA model is significantly faster for each estimation model run. We also need to consider that the LSTM model was run on ten different epoch values five times, resulting in 50 runs to identify the best model fit. Therefore, the time for LSTM is significantly higher, but results in significantly better predictions. However, when looking at the number of epochs, we ran on more epochs than we needed. We could have easily reduced the number of epochs to 4 instead of 10. Nevertheless, for this study, it was important to identify suitable epochs, so that future runs target a minimal number of them. We also could have terminated runs that showed a non-promising loss function in contrast to previous experiments.

**Table 3.** Runtime comparison 5950X.

| Cryptocurrency | LSTM | ARIMA |
|---|---|---|
| EOS | 186.24 s | 39.03 s |
| BTC | 356.79 s | 27.19 s |
| ETH | 215.48 s | 33.47 s |
| DOGE | 216.00 s | 83.53 s |

## 8. Conclusions

Our model provides as an excellent base model to add further input parameters such as the *High*, *Low*, and *Volume* data of the cryptocurrencies, as well as being able to add other hyperparameters and changes to the model layers. Our LSTM approach produces a more accurate RMSE than the ARIMA model at the cost of a longer runtime.

The prediction lines had a minimal deviation from the actual recorded close values. Additionally, the model has very high accuracy as the predictions had a minimal root-mean-squared error, meaning the predicted value was close to the actual price. The model must be hyperparametrized to account for variables such as investor sentiment. Furthermore, our naive approach of only one-day prediction, based on regular time series, does not produce a high-quality result with an expedient response because our response has a one-day delay.

The managerial implications of our findings include the possibility of creating a product for investors, who can build upon our model to add other hyperparameters, creating an even more accurate model to predict cryptocurrency price. The limitations of our paper are that we have not explored the architecture of introducing such hyperparameters in a detailed manner.

For future research, the model can accept a hyperparameter such as sentiment analysis of tweets from Twitter pages or even simply the measurement of tweet volume.

**Author Contributions:** Conceptualization, J.P.F. and G.v.L.; methodology, J.P.F. and G.v.L.; validation, J.P.F. and G.v.L.; formal analysis, J.P.F. and G.v.L.; investigation, J.P.F. and G.v.L.; data curation, J.P.F. and G.v.L.; writing—original draft preparation, J.P.F. and G.v.L.; writing—review and editing, J.P.F., G.v.L., C.T. and Y.J.P.B.; visualization, J.P.F. and G.v.L.; funding acquisition, G.v.L.; providing support for the initial education on data science and LSTM, C.T. and Y.J.P.B. All authors have read and agreed to the published version of the manuscript.

**Funding:** Work by G.v.L. was in part supported by the NSF Grant #1829704: CyberTraining: CIC: CyberTraining for Students and Technologies from Generation Z and the NIST Transfer award 60NANB21D151 executed at the University of Virginia in support of the Reusable Hybrid and Multi-Cloud Analytics Service Framework. Work by J.P.F. was supported by the NSF Grant titled *Florida Georgia Louis Stokes Alliance for Minority Participation* HRD 1201981 with subcontract to Florida A&M University (FAMU C-5083).

**Institutional Review Board Statement:** Not applicable.

**Informed Consent Statement:** Not applicable.

**Data Availability Statement:** Publicly accessible data from Yahoo Finance were used [14].

**Conflicts of Interest:** The authors declare no conflict of interest.

**Sample Availability:** The following additional material is available: (a) an early draft paper [21] with limited data and incomplete analysis and (b) the program located on GitHub [18].

## Abbreviations

The following abbreviations are used in this manuscript:

| | |
|---|---|
| MDPI | Multidisciplinary Digital Publishing Institute |
| LSTM | Long short-term memory |
| AI | Artificial intelligence |
| EOS | Electro-Optical System |
| DApps | Decentralized apps |
| USD | United States Dollar |
| API | Application programming interface |
| RMSE | Root-mean-squared error |
| GTX | Giga Texel Shader eXtreme |
| RTX | Ray Tracing Texel eXtreme |
| BTC | Bitcoin |
| ETH | Ethereum |
| LTS | Long-term support |
| ARIMA | Autoregressive integrated moving average |

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
