# Peer review of "Time Series Analysis of Cryptocurrency Prices Using Long Short-Term Memory"

_algorithms, doi:10.3390/a15070230_

Round 1
Reviewer 1 Report
Dear Author(s),
Please find below my concerns and recommendations regarding your manuscript proposal entitled "Time Series Analysis of Blockchain-Based Cryptocurrency Price Changes"
1. First of all, during my initial documentation for this review, I found that your manuscript has many similarities with the articles and resources available at the following addresses:
https://arxiv.org/abs/2202.13874?cv=1
https://github.com/cybertraining-dsc/su21-reu-361/blob/main/project/index.md?cv=1
Please clarify this important issue, because it is supposed that now you intend to publish a new and original research article in the Algorithms Journal.
2. The Introduction section should be improved, by defining and describing:
- the research gap:
- the research goal;
- the research question(s).
The research articles should include all these elements, so that the research framework be clearly defined.
3.1. The Introduction section must be improved by including additional references.
3.2. Also, the article should have an extended Literature Review section where you present the previous results from the scientific literature. Based on the gap(s) identified in the literature, I recommend you to define and describe your research hypothesis. This hypothesis should be tested so that you can validate or invalidate it.
4. The DATASETS section should explain the composition of your used datasets. The details are useful for the readers.
5. The ARCHITECTURE section must be improved. Please specify if the architecture is your own proposal or it is a validated one from the previous scientific literature.
6. The BENCHMARK section should also include a comparative study with the existing results from previous models.
7. In the CONCLUSIONS chapter, please describe the following important aspects:
- your main findings;
- the research limitations;
- the managerial implications (here is the place where you can "sell" your research results to the readers).
Dear Author(s),
Please consider all the above remarks as being constructive recommendations in order to improve the general quality of your manuscript proposal.
Kind Regards!
--------------
Reviewer 2 Report
Dear Authors,
Here are my comments:
1. Introduction section is thin.
2. Literature review section is missing. However the literature on cryptocurrency price change is rich... Please make it a comprehensive section covering ... Previously what methods are deployed to study the cryptocurrency price change... and why one should use AI and neural network to study the cryptocurrency price change. In this regard authors may consider these and/or similar studies:
Cryptocurrencies chaotic co‐movement forecasting with neural networks. Internet Technology Letters, 3(3), e157
Bitcoin price change and trend prediction through twitter sentiment and data volume. Financial Innovation, 8(1), 1-20.
COVID-19 pandemic improves market signals of cryptocurrencies–evidence from Bitcoin, Bitcoin Cash, Ethereum, and Litecoin. Finance Research Letters, 44, 102049.
3. Authors quoted "The experiment is further confounded by the nature of stock prices: they follow random walk theory" ... On what basis authors have arrived to this conclusion... did they use any specific test etc.
4. Every section need more detailed discussions for the ease of the readers.
Best.
Round 2
Reviewer 1 Report
Dear Author(s),
I have read the revised version of your manuscript proposal and I saw that the new proposal is completely different from the previous version.
Please see below my remarks and recommendations:
- the Introduction section is well-written, but you need to specify the research question(s). By presenting the research question(s), the readers will understand from the very beginning what you want to cover by your research proposal.
- at the end of the "Literature Review" chapter you should have a clear description of the research hypothesis. Every modern scientific article must have at least one research hypothesis. I tried to find the research hypothesis in your article, but I didn't see it.
- within sections "3. Datasets" and "4. Architecture" and "5. Implementation" you shortly present the data and the implementation. I warmly recommend you to include relevant pieces of code (in your case: Python). Readers need to understand and see the core of the process.
- in chapter 6 you present the benchmark. Is it also valid for the latest/recent evolutions of Bitcoin and other crypto-currencies?!?
- in the Conclusion section you should present: the managerial implications and the limitations of you research
- the list of references should be improved.
Kind Regards!
Author Response
I have read the revised version of your manuscript proposal and I saw
that the new proposal is completely different from the previous
version.
ANSWER:
* Thank you, the original review stated we need to significantly
change the paper and distinguish it from our previous work,
which we have done.
Please see below my remarks and recommendations:
- the Introduction section is well-written, but you need to specify
the research question(s). By presenting the research question(s),
the readers will understand from the very beginning what you want to
cover by your research proposal.
ANSWER:
* The research question is now specified at the end of the Introduction section.
- at the end of the "Literature Review" chapter you should have a
clear description of the research hypothesis. Every modern
scientific article must have at least one research hypothesis. I
tried to find the research hypothesis in your article, but I didn't
see it.
ANSWER:
* The hypothesis is now included at the end of the Literature Review chapter.
- within sections "3. Datasets" and "4. Architecture" and
"5. Implementation" you shortly present the data and the
implementation. I warmly recommend you to include relevant pieces of
code (in your case: Python). Readers need to understand and see the
core of the process.
ANSWER:
* There is now a figure within the implementation section that showcases a diagram of the layers of the LSTM. Additionally, there is a new Python pseudocode figure that describes the code of the program.
- in chapter 6 you present the benchmark. Is it also valid for the
latest/recent evolutions of Bitcoin and other crypto-currencies?!?
ANSWER:
* The time period that we consider already includes periods of high volatility. If you refer to the recent collapse of Bitcoin, such a timeframe resides outside of what we examine within this paper. Incorporating this recent event would change the original hypothesis of the paper— we would have to write a new paper to look more in detail on this particular period and its predictability.
- in the Conclusion section you should present: the managerial
implications and the limitations of you research
ANSWER:
* The managerial implications and limitations are now included in the Conclusion section.
- the list of references should be improved.
ANSWER:
* We have added additional reputable sources that further back our observation that LSTM is a well-explored methodology for forecasting.
Reviewer 2 Report
Good Luck !!!
Author Response
Thank You!
Round 3
Reviewer 1 Report
Dear Author(s),
I consider you addressed all my constructive recommendations from the previous round of review. Now I have one minor remark: the figures 11, 12, 13, 14, 15, 16 are located between references from the final bibliography: reference 16 and reference 17.
Please revise this issue.
Kind Regards!
Author Response
Greetings
Document was reviewed and the figures have been placed before the references. In addition, the document was double-checked by native English speakers for grammar, spelling, and conciseness. Thank you kindly.